# Targeting the Brain with Single-Domain Antibodies: Greater Potential Than Stated So Far?

**DOI:** 10.3390/ijms24032632

**Published:** 2023-01-30

**Authors:** Mireille Elodie Tsitokana, Pierre-André Lafon, Laurent Prézeau, Jean-Philippe Pin, Philippe Rondard

**Affiliations:** Institute of Functional Genomics, University of Montpellier, CNRS, INSERM, 34094 Montpellier, France

**Keywords:** camelid nanobody, VHH, blood-brain barrier, immunotherapy, brain penetration

## Abstract

Treatments for central nervous system diseases with therapeutic antibodies have been increasingly investigated over the last decades, leading to some approved monoclonal antibodies for brain disease therapies. The detection of biomarkers for diagnosis purposes with non-invasive antibody-based imaging approaches has also been explored in brain cancers. However, antibodies generally display a low capability of reaching the brain, as they do not efficiently cross the blood−brain barrier. As an alternative, recent studies have focused on single-domain antibodies (sdAbs) that correspond to the antigen-binding fragment. While some reports indicate that the brain uptake of these small antibodies is still low, the number of studies reporting brain-penetrating sdAbs is increasing. In this review, we provide an overview of methods used to assess or evaluate brain penetration of sdAbs and discuss the pros and cons that could affect the identification of brain-penetrating sdAbs of therapeutic or diagnostic interest.

## 1. Introduction

The use of monoclonal antibodies (mAbs) is one of the most significant advances of the last decades in medicine. The effectiveness of therapeutic antibodies has been observed in oncology and inflammation, where they have become a dominant drug-developing pipeline [1]. Nevertheless, their development in neurology is only at its beginning. This delay can be explained by the complexity of brain diseases and by the existence of anatomical physical barriers, mainly the blood-brain barrier (BBB) [2,3], hiding behind therapeutic targets. Access to the brain by antibodies, which are high molecular weight molecules, is limited by this BBB [4,5]. Thus, the use of antibody fragments represents a good alternative. Several reviews have already reported the advantageous features that single-domain antibodies (sdAbs) possess over traditional antibodies in terms of size—they are 10-fold smaller (2.5 nm × 4 nm, 12–15 kDa)—also in terms of cost of production, conformational specificity, and engineering [6,7,8,9,10,11]. The use of single-domain antibodies VHH (also called Nanobody^®^, Ablynx Sanofi, Ghent, Belgium), VNAR (variable domain of new antigen receptor), and VH were also highlighted in the treatment of central nervous pathologies. VHH [12] and VNAR [13] correspond to the smallest antigen binding domain derived from the heavy-chain-only antibody naturally present in camelids and sharks, while VH is from conventional antibodies present in mammals. Their molecular features and functional mechanisms have been well described in the literature [14,15,16,17,18,19]. If there are more and more reports on sdAb engineering in order to reach the brain through different mechanisms, there is a need for better approaches to evaluate and rapidly identify a lead sdAb with a better starting probability of reaching clinical trials.

In this review, we aim first to illustrate the progressive use of mAbs in neurodegenerative diseases and neuro-oncology therapy and the promising applications of sdAbs in the same field. Second, we will describe the methods used to assess the ability of the sdAbs to enter the brain parenchyma, with a special emphasis on new rapid and efficient technical approaches to accelerate and improve the screening and development of single-chain domains.

## 2. Application of Antibodies in Brain Diseases

In seeking new therapies to treat neurological, neurodegenerative, and psychiatric diseases, small molecules have been the drugs of choice for many years. However, while lots of them display efficient beneficial action, they often display strong secondary side effects that counteract the benefits or the health of the patients. Off-targets, lack of selectivity, tolerance, and receptor desensitization are mechanisms reported for their side effects [20,21]. Antibodies and nanobodies have then emerged as promising therapeutic biomolecules, thanks to their high affinity to their target and their high selectivity. They are now developed for treating more and more neurological and neurodegenerative diseases. Due to their convex paratope, nanobodies share a propensity for recognizing cryptic allosteric epitopes (sites topographically different from where the endogenous ligand binds). Therefore, they are particularly prone to recognize specific conformations, stabilize them, distinguish between homo and heterodimers [10,22,23,24], and exert a fine allosteric modulation of the receptors’ activity in the presence of the endogenous ligand. Recently, a lot of effort in drug discovery programs focused on allostery to modulate neuroreceptors for the treatment of central nervous system (CNS) disorders [25]. Thus, although there could be a limitation for both antibodies and nanobodies because of their low ability to penetrate the brain by themselves, nanobodies emerge as a more promising tool compared with mAbs. Note that most of the mAbs that will be cited thereafter exert their pharmacological effects outside the brain or in regions the BBB does not protect.

### 2.1. Multiple Sclerosis

The first monoclonal antibody, called natalizumab (Tysabri^®^, Biogen Idec/ELAN), was approved for a neurological indication in 2004 by the United States Food and Drug Administration (FDA) and in 2009 by the European Medicines Agency (EMA) (Table 1). It is indicated for the treatment of multiple sclerosis (MS), a neurodegenerative disease of the CNS in which inflammation and autoimmunity are involved. Natalizumab prevents lymphocyte transport across the BBB by blocking the binding of α4β1 integrin (or VLA-4 for very late antigen-4) present on the surface of T cells to the adhesion molecule VCAM (vascular cell adhesion protein) expressed by endothelial cells in the BBB. Thereby, it plays an anti-inflammatory role in the CNS [26]. Used off-label, another monoclonal antibody, rituximab (Rituxan^TM^, phase III trial), showed efficacy in reducing recurrences by lysing circulating B cells via binding to CD20 [27,28,29]. Since then, several mAbs have been developed both for relapsing and primary progressive forms of MS or for repairing damage: ocrelizumab (Ocrevus^TM^, approved in 2017) [30], ofatumumab (approved in 2020), ublituximab (in phase III studies, pending FDA approval) [31], alemtuzumab (Lemtrada^TM^, approved in 2014) [32], opicinumab (BIIB033, phase II trials) [33], elezanumab (ABT-555, in phase II trials) [34], and temelimab (GNbAC1, phase II studies) [27,35] (Table 1). Unlike the majority, opicinumab aims at penetrating the brain parenchyma to act like an antagonistic antibody on LINGO-1, a transmembrane cell-surface glycoprotein expressed on neurons and oligodendrocytes [36]. It reached phase II studies [37,38], but its development has been stopped due to its failure to improve patient conditions [39]. Nevertheless, mAbs have become the preferred therapy for MS [27], and this disease remained for a long time the only pathology for which they were used in the field of neurology.

### 2.2. Migraine

The use of mAbs in treating neurological conditions has expanded to migraine. Due to its severity and its high prevalence worldwide (15–18% in a year) [41,42], migraine is the most disabling neurological disorder [43]. It triggers a huge economic cost to society, which is estimated to be billions of dollars or euros per year in the United States and the European Union, respectively [44,45]. This financial burden derives from the cost of medical care, absence from work, and reduced productivity.

In the hope of proposing an effective therapy, researchers have investigated the pathophysiology of migraine for several centuries. It led to two main theories—one attributing migraine pain to vasodilatation of cerebral arteries and meningeal arteries (also called “the vascular theory”); and the other one, to neural disorders of the CNS (also called ‘the central neuronal theory’) [46]. While there is still a debate on these two opposing hypotheses, research progress permitted the identification of neuropeptide calcitonin gene-related protein (CGRP) and its receptors in the trigeminal ganglion and the paraventricular structures as a key player in the disease [47]. Studies revealed that the level of CGRP increases during migraine attacks. Besides, its administration for migraine patients triggers headaches, while no effect was observed in healthy volunteers. Experimental results also suggest that CGRP is a potent vasodilator and can modulate pain by enhancing neurotransmission in the migraine circuit [46,48]. This discovery has offered new possibilities to overcome the limited efficacy of the available drugs with the arrival of four mAbs blocking the CGRP receptor or ligand [47]. Erenumab (Aimovig^®^), fremanezumab (Ajovy^TM^), and galcanezumab (Emgality^®^) received regulatory approval as prophylaxis treatment in 2018, and eptinezumab (Vyepty^TM^) in 2020 [49,50,51,52,53,54,55,56] (Table 2). These therapies demonstrated efficacy in phase II and III trials by reducing the frequency of migraine attacks and improving patients’ ability to carry out daily activities [57]. However, a cure therapy is still awaited.

CGRP antagonism requires clinical vigilance, notably because of the ubiquitous feature of CGRP and its receptors. Indeed, they are expressed throughout the body, not only in the trigeminal system. Thus, off-targets are possible, and this may engender inflammatory complications, as reported in 2021 by clinical case series [58]. To deal with this issue, we can think of using sdAbs. With their rapid elimination rate from the systemic circulation, a lesser accumulation in peripheral organs could limit their binding to off-targets. However, this must go along with rapid and prolonged CNS distribution. CNS-directed sdAbs are likely needed. This approach may require a lot of engineering. That may explain why, to our knowledge, no sdAb has been tested so far for this pathology.

### 2.3. Brain Tumors

The success of mAbs against many cancers has naturally led to their use in brain tumors such as glioblastoma, primary central nervous system lymphoma (PCNSL), and brain metastases. Glioblastoma is the most common and deadliest type of primary brain tumor (a tumor that arises within the brain). It develops from glial cells or their precursors. Along with its growth, it induces a local formation of new vasculature characterized by a high expression of vascular endothelial growth factor (VEGF). The approach used to slow down its progression is to deprive tumoral cells of their increasing blood supply by targeting circulating VEGF and thus blocking its binding to the VEGF receptor with bevacizumab (Avastin^®^) [59,60] (Table 3).

However, the efficacy of angiogenesis inhibitors is insufficient to prolong survival, and patients experience tumor relapses. This could be explained by the diffusive progression of angiogenesis-independent infiltrating tumor cells thriving on matured vessels. Therefore, strategies to starve cancer cells by directly targeting vasculature in the brain tumor have been investigated. To identify VHHs selective for tumor vessels, a VHH displaying phage library was administered intravenously (i.v.) to mice bearing orthotopic E98 human glioma xenografts. This led to the identification of VHH C-C7, which selectively recognizes a subpopulation of tumor vessels [61] (Table 3). However, the pharmacological activity of C-C7 on the tumor progression remains to be assessed, as well as its brain penetrance ability.

Glioblastoma stem-like cells (GSC) are hypothesized to trigger therapy resistance and tumor relapse. They contribute to poor patient survival. VH-9.7, a VH fragment, has been identified to selectively target GSC [62,63] (Table 3). It recognizes five patient-derived GSC lines, which are named 12.1 GSC (generating focal tumor xenograft in NOD-SCID mice), 22 GSC, 33 GSC (minimally invasive), 44 GSC, and 99 GSC (highly infiltrative). Unlike with an immunoglobulin G (IgG) and a single-chain variable fragment (scFv), a local tumor accumulation was observed by infrared spectroscopy 30 min after an intravenous administration of VH-9.7 (300 pmol) in a mouse model carrying a 22 GSC intracerebral xenograft [63]. This opens the field to diagnostic and therapeutic perspectives of this VH (patent WO 2019/074892 A1) [62].

In the case of PCNSL, the monoclonal antibody rituximab (Rituxan™) is incorporated into treatment regimens [64,65] to block B lymphocytes’ extravasation following the mechanism described in the multiple sclerosis section (Table 3).

Anti-HER2 (epidermal growth factor receptor-2) trastuzumab (Herceptin^®^) has been tested in brain metastasis associated with HER-2-positive breast cancer. It could delay the onset of brain symptoms; however, it cannot stop the tumor progression in the brain [66].

**Table 3 ijms-24-02632-t003:** Antibodies under investigation or approved for brain tumors.

Antibody	Target	Clinical Status	Dose	Key Findings/Mode of Action	References
Bevacizumab (Avastin^®^) *Humanized IgG1 mAb*	VEGF	Approved by FDA and EMA	i.v. injection 15 mg/kg every 3 weeks In combination with other antibodies	Binds to circulating VEGF and inhibits its binding to VEGFR	[59,60]
C-C7 *sdAb*	Dynactin-1-p150^Glued^	Preclinical stages	/	Targets selectively a subpopulation of tumor vessels	[61]
VH-9.7 *VH fragment*	Human GSC xenografts. Specific target unknown	Preclinical stages	i.v. injection of 300 pmol in a mouse model with intracerebral xenograft	Localizes local tumor accumulation	[63]
Rituximab (Rituxan™) *Chimeric mAb*	CD20	Phase III	Infusion of 500 or 1000 mg every 6–12 months	Blocks B lymphocytes’ extravasation	[64,65]
Trastuzumab (Herceptin^®^) *Humanized IgG1 mAb*	HER2 receptor	Approved by FDA and EMA	Loading dose of 8 mg/kg and 6 mg/kg every 3 weeks for 52 weeks (infusion)	Delays the onset of brain symptoms of metastasis from breast cancer	[66]

### 2.4. Alzheimer’s Disease

Alzheimer’s disease (AD) is the most common neurodegenerative disorder, contributing to 60–70% of dementia cases [67,68]. It is characterized by two major brain changes: accumulation of amyloid-β (Aβ) peptides leading to extracellular senile plaques and accumulation of intracellular hyper-phosphorylated Tau proteins called neurofibrillary tangles (NFT) [69,70]. Accumulation of both protein markers in the brain leads to progressive memory loss and, ultimately, to dementia. The pathology is multifactorial, and the underlying mechanisms involved in its progression are not yet fully elucidated. Current drugs approved for its treatment are only symptomatic, have modest benefits, and do not prevent the progression of the pathology [69,70]. In 2018, the French National Authority for Health stopped reimbursing the four main drugs available on the market (Memantine^®^, Donepezil^®^, Galantamine^®^, and Rivastigmine^®^), arguing that these treatments present insufficient medical interest to justify their reimbursement.

For the last 25 years, therapeutic research on AD has focused on developing Aβ-targeting drugs, but they have mainly failed to demonstrate clinical efficacy [71]. Thus, there is still an enormous need to develop therapies with the potential to slow down or stop AD progression. A promising breakthrough in reaching these objectives has been the use of mAbs. Among the different strategies tested, six mAbs have reached phase III trials: bapineuzumab (AAB-001) [72], solanezumab (LY2062430) [73], crenezumab (MABT5102A) [74,75], gantenerumab (RO4909832) [76,77], aducanumab (Alduhem^TM^) [78,79], and lecanemab (BAN2401) [80] (Table 4). However, the early excitement was quickly replaced by disappointment when it became clear that most of them did not exhibit the expected great success [70]. While research on bapineuzumab was terminated due to safety concerns [81,82,83], clinical trials with solanezumab [73] and crenezumab [84] were stopped because of the absence of cognitive or functional improvement. Concerning gantenerumab, a phase III clinical trial is still ongoing (NCT03444870, NCT03443973, NCT03887455) [85,86,87]. To date, aducanumab (Aduhelm^TM^) [88] obtained controversial approval from the FDA [89] but was rejected by the EMA. This provisional regulatory decision was taken under an accelerated approval pathway. It requires the validation of clinical benefit in a post-approval trial to be held [90,91]. Aducanumab was demonstrated to reduce both Aβ and Tau aggregates [88,92,93] and to improve cognitive deficits in the early stages of the pathology [89], representing a hope to slow down the progression of AD. However, repeated injections and high doses have reported incidents of amyloid-related imaging abnormalities (ARIA), such as micro-hemorrhages and oedemas, in 43% of patients treated with aducanumab [92]. These ARIAs are due to the production of anti-aducanumab antibodies in the blood and cerebrospinal fluid of AD patients, reflecting an immunogenic response towards these IgGs. More recently, lecanemab was approved by the FDA in January 2023 [94].

Anti-Tau therapies have also been explored, and four mAbs reached phase II trials: gosuranemab (BIIB092) [95], tilavonemab (ABBV-8E12) [96], semorinemab (RO7105705) [97,98], and zagotenemab (LY3303560) [70] (Table 4). The failure of gosuranemab [95], tilavonemab [99], and zagotenemab [100] has recently been reported (Table 4).

To possibly overcome the immunogenic side effects of therapeutic mAbs, sdAbs could constitute a real breakthrough for the treatment of AD. Indeed, VHHs are supposed to have a low or non-existent capacity to produce immunogenicity reactions after injection [101]. A dozen sdAbs have shown their potential therapeutic or diagnostic value for AD in vitro [102]. We can cite two of them which reached in vivo investigations: R3VQ and A2 that bind brain Aβ deposits and Tau inclusions, respectively [103]. Staining throughout the brain was observed in PS2APP mice (a genetic model harboring β-amyloid lesions) 4 h after intravenous administration of 50 mg/kg of fluorescently labeled R3VQ (noted R3VQ-S-AF488). In contrast, only a few plaques were positively stained with a fluorescently labeled IgG, suggesting that VHH could better penetrate the brain and more easily reach its target (Table 4). The intensity of the labeling was reduced when using a lower dose of R3VQ-S-AF488 (10 mg/kg). Extravasation of R3VQ-S-AF488 from blood to brain parenchyma was observed in vivo using transcranial two-photon laser scanning microscopy on the superficial cortex. Amyloid deposits started to be visualized 30 min post-intravenous injection and stayed up to 4 h post-injection [103]. In wild-type mice, extravasation of R3VQ-S-AF488 was also observed by administrating 50 mg/kg via the intravenous route but then rapidly vanished—likely because it was not retained in the brain in the absence of its cerebral target. VHH A2, on the other hand, labels NFT-like structures within neurons of Tg4510 mice (a genetic model bearing NFT) following an intravenous administration (10 mg/kg). Real-time imaging showed a delay in the staining of this NFT-like structure that can be explained by the intracellular localization of the target. No labeling was observed in control animals: wild-type mice and mice that received a conventional anti-phosphorylated Tau antibody [103] (Table 4). Even though these sdAbs, targeting Aβ and hyperphosphorylated Tau, are still barely studied, their preclinical development could constitute a real breakthrough in the immunotherapy field.

**Table 4 ijms-24-02632-t004:** Nonexhaustive list of antibodies tested in Alzheimer’s disease therapy.

Antibody	Target	Clinical Status	Dose	Key Findings/Mode of Action	References
Bapineuzumab (AAB-001) *Humanized IgG1 mAb*	N-terminal region of Aβ Targets fibrillar and soluble monomeric forms	Phase III—discontinued	0.5, 1.5, or 5 mg/kg through i.v. injection, every 13 weeks for 78 weeks	No differences observed compared to placebo groups. Slight clearance of fibrillar cerebral Aβ	[72,81,82,83]
Solanezumab (LY2062430) *Humanized IgG1 mAb*	Mid-domain of Aβ Targets soluble monomeric forms	Phase III—discontinued	i.v. infusion of 400 mg once a month for 80 weeks	No reduced cognitive decline in mild AD patients versus placebo	[73]
Crenezumab (MABT5102A) *Humanized IgG4 mAb*	Aggregated Aβ forms (oligomeric, fibrillar and plaques)	Phase III—discontinued	i.v. infusion of 60 mg/kg every 4 weeks for 100 weeks	Lowering of 20% of cognitive decline. Stabilization of Aβ42 and rise of Aβ40 levels	[74,75]
Gantenerumab (RO4909832) *Human IgG1 mAb*	N-terminal and mid-domain of Aβ. Targets Aβ fibrils	Phase III	s.c. injection of 255 mg or 510 mg every week	Reduces Aβ plaques, CSF total Tau, and phospho-Tau181	[76,77]
Aducanumab (Alduhem^TM^) *Human IgG1 mAb*	Aggregated Aβ forms	Approved by FDA—Phase IV confirmatory trial ongoing	Monthly i.v. infusion. Titrated dosing reaching 10 mg/kg at the 7th infusion	Reduction in cognitive decline, Aβ and Tau levels	[78,79,88,89,92,93]
Lecanemab (BAN2401) *Humanized IgG1 mAb*	Large and soluble Aβ protofibrils	Approved by FDA	i.v. infusion of 5 mg/kg or 10 mg/kg every 2 or 4 weeks for 18 months	Reduction of brain Aβ and decreased cognitive decline	[80,94]
Gosuranemab (BIIB092) *Humanized IgG4 mAb*	N-terminal domain of Tau	Phase II—discontinued	i.v. infusion of 2100 mg every 4 weeks for 1.5 years	Increased cognitive decline	[95]
Tilavonemab (ABBV-8E12) *Humanized IgG4 mAb*	N-terminal domain of extracellular aggregated Tau	Phase II—discontinued	i.v. infusion of 2000 or 4000 mg at days 1, 15 and 29, and every 28 days for 1 year	No benefit of antibody over placebo	[96,99]
Semorinemab (RO7105705) *Humanized IgG4 mAb*	Extracellular aggregated Tau	Phase II	i.v. infusion of 1500 mg, 4500 mg, or 8100 mg every 2 weeks for the 3 first infusions, and every 4 weeks for 73 weeks	No changes observed between antibody-treated and placebo patients	[97,98]
Zagotenemab (LY3303560) *Humanized mAb*	N-terminal domain of Tau	Phase II—discontinued	i.v. infusion of 1400 or 5600 mg every 4 weeks for 100 weeks	No benefit of antibody over placebo	[70]
R3VQ *sdAb*	Aβ brain aggregates	Preclinical stages	i.v. infusion of 50 mg/kg to PS2APP mouse model	Crosses the BBB and binds to amyloid aggregates	[103]
A2 *sdAb*	Tau inclusions	Preclinical stages	i.v. infusion of 10 mg/kg to Tg4510 mouse model	Crosses the BBB and binds to NFTs	[103]

### 2.5. Parkinson’s Disease

After AD, Parkinson’s disease (PD) is the most prevalent neurodegenerative disorder. It is characterized by a loss in dopamine-producing neurons in the substantia nigra (in the midbrain). The pathology is associated, in most cases, with intracellular inclusions—called Lewy bodies [104]—containing pathogenic α-synuclein protein abnormally folded [105,106]. Aggregates of α-synuclein are also observed abundantly in the extracellular space of the pathological environment [107]. These oligomers are thought to mediate the propagation of the disease by neuron-to-neuron transmission [107]. To halt PD progression, several antibodies against the oligomeric form of α-synuclein have been tested [106] (Table 5): prasinezumab (PRX002), cinpanemab (BIIB054) [108], MEDI1341, and a dozen of other antibodies that, so far, have been tested in preclinical studies only. Prasinezumab was the first to undergo clinical studies in 2017. Phase I and 1b trials in healthy and mild idiopathic PD patients treated with prasinezumab showed favorable results regarding safety, tolerability, and pharmacokinetics. The treatment reduced the level of free α-synuclein in the blood. Besides, no serious adverse effects nor immunogenicity were observed [109,110]. However, the results of the phase II trial (NCT03100149) are less positive [111]. No significant changes were seen in the symptoms of PD nor the imaging measures of the disease’s progression. However, despite this failure, studies on safety and efficacy in early-stage PD patients are continuing (phase 2b; NCT04777331) [112]. Cinpanemab was also reported as negative after a phase II trial (NCT03318523) [108]. Several adverse effects were observed and led its developer to discontinue the study [108]. Regarding the third antibody MEDI1341, it is currently in a phase I trial (NCT04449484) [113].

In the same idea of targeting α-synuclein aggregates, VHHs have been generated; however, the strategy is different. Here, the authors try to target intracellular α-synuclein oligomers [114]. Moreover, to do so, they use genetic material to express the VHHs directly in the cytoplasm. Since this review focuses on purified sdAbs administered extracellularly, we will not develop this administration technique for antibody fragments. However, it is interesting to know that the VHH PFFNB2, for example, specifically binds to α-synuclein preformed fibrils and does not recognize the monomers in vitro. It can also significantly dissociate the fibrils. Its expression in animals prevented the spreading of the α-synuclein pathology to the cortex [114] (Table 5). Other studies have biochemically well-characterized NbSyn2 [115] and NbSyn87 [116]. They have also assessed their therapeutic [117,118] and diagnostic potentials [119]. These two VHHs recognize monomeric and fibrillar forms of α-synuclein. They were found to inhibit the formation of fibrils and convert toxic oligomers into less toxic species [120] (Table 5). These effects dramatically reduced the oligomers’ toxicity in cell lines overexpressing α-synuclein.

PD has long been considered a non-genetic disorder. However, genetics research has revealed that at least nine genes are incriminated in 5–10% of patients with the monogenic form of the pathology [121]. One can cite the gene encoding for leucine-rich-repeat kinase 2 (LRRK2). Mutations in LRRK2 are among the most common causes of inherited PD, while an overactivation of LRRK2 has also been associated with the more frequent idiopathic form of the pathology. Following these observations, its inhibition appears to be an appealing approach for drug development. In 2022, Singh and colleagues reported the identification of seven VHHs acting as a negative allosteric modulator of LRRK2 [122]. Their modulation activity is wide. Three of them exert a total inhibition of the tested kinase activities of LRRK2: autophosphorylation and Rab phosphorylation in cells, as well as LRRK2 peptide substrate and Rab phosphorylation using a commercialized assay (PhosphoSens Protein Kinase Assay). Three others only inhibit Rab phosphorylation in cells and with the protein kinase assay, while the last one only acts on autophosphorylation and Rab phosphorylation in cells. Further investigation into their therapeutic use is ongoing.

### 2.6. Creutzfeldt-Jacob Disease

Creutzfeldt-Jacob’s disease (CJD) is associated with the conversion of the cellular prion protein (PrP^C^), rich in alpha helices, into a beta-rich structure conformer, the PrP^Sc^ [123]. The use of antibody fragments as diagnostic and therapeutic tools has grown in interest over the years [102,124,125]. It is not surprising that their application has been tested in protein-misfolding diseases with a rapid fatal fate, such as CJD’s disease. Despite the disappointing outcomes of most immunotherapies tested on other neurodegenerative diseases, CJD and prion diseases, in general, have unique features. Indeed, PrP^Sc^ is a well-characterized causing agent [126,127], which makes it a highly valid therapeutic target. In addition, the conversion of the PrP^C^ into PrP^Sc^ takes place at the cell surface [128], making these targets easily reachable by antibodies. Several mAbs have been produced and tested in vitro and in vivo [129,130,131], but only one mAb was tested on CJD patients [132]. Indeed, a humanized anti-PrP^C^ monoclonal antibody (PRN100), able to reach the brain, has been used to treat six patients with CJD and administered intravenously [133] (Table 6). No significant adverse outcomes have been noticed during the treatment of the patients. Even if the progression of the disease has not been halted or reversed, neuropathological examinations revealed modifications of the PrP^Sc^ deposition in brains [133]. These encouraging results with the PRN100 will now need to be evaluated in the phase II clinical study with patients enrolled at the earliest clinical stages.

In parallel, a VHH, PrioV3, targeting the PrP^C^, is currently being developed at a preclinical stage [134]. Its brain uptake ability was assessed in adult wild-type FVB/N mice following an intraperitoneal administration (100 µg per mouse representing an approximative dose of 5 mg/kg). Immunodetection of PrioV3 showed a biphasic pattern in brain homogenates. It peaked at 12 h before resurging at 72 h, while no signal was detected in the saline-treated mice. Its brain distribution showed an accumulation in the hippocampus and alveus at 4 to 12 h and in the cerebellar cortex after 24 h post-injection [134] (Table 6 and Table 7). Nevertheless, the efficacy of PrioV3 in abrogating prion propagation in infected mouse brains is lacking in the mentioned study. However, a decreased prion replication in the spleen was observed.

## 3. Methods to Assess Brain Penetration of sdAbs

Many studies have demonstrated the ability of sdAbs to reach deep brain tissue compared to conventional antibodies [103]. They brought to light well-known VHHs and VNARs, which became carriers for biologics to increase BBB transport upon peripheral administration. One can cite E9, FC5, TXB4, and IGF1R5 that target the intracellular GFAP, the extracellular TMEM-30A, transferrin receptor, and insulin-like growth factor 1 receptor (IGF-1R), respectively [125] (Table 7). While their transport across the BBB is attributed to a receptor or adsorptive-mediated transcytosis, some additional brain-penetrating sdAbs identified can reach the brain by an unclear mechanism. Unlike the dogma, sdAbs seem to have great potential in crossing the BBB. Pursuing the development of sdAbs into clinical stages for brain pathologies requires validating their brain penetration ability after a peripheral administration. Here we review the methods used to do so, either directly or indirectly, and discuss their pros and cons.

### 3.1. Transmigration across In Vitro BBB Models

The exchange of molecules between blood and the brain is tightly and mainly controlled by the BBB located within the cerebral microvessels. Briefly, the BBB is made up of a layer of endothelial cells surrounded by astrocytic endfeet and pericytes. The whole is called a neurovascular unit. It plays a physiological role in maintaining the proper functioning of the brain. However, it hampers the delivery of neuropharmaceuticals. In 2002, Muruganandam and colleagues studied the transmigration of sdAbs across an in vitro model of the human BBB obtained by seeding human cerebromicroventricular endothelial cells (HCEC) onto a porous membrane (1 µm-size pores) (Figure 1A). Below, a second chamber containing medium conditioned by fetal-human astrocytes was used to induce a BBB-like phenotype. The ability of soluble cMyc-His tagged sdAbs to transmigrate was then assessed by adding 100 µg of sdAbs into this monolayer of endothelial cells and measuring the concentration obtained in the bottom chamber by using enzyme-linked immunosorbent assay (ELISA). This BBB model is impermeable to 10 kDa-dextran, as observed when assessing BBB integrity. This approach resulted in the identification of FC5 and FC44, two distinct ~14 kDa-camelid sdAbs recognizing HCEC (Table 7). These two can transmigrate across this human in vitro BBB by a transcellular passage [135].

Transport of FC5 and FC44 has also been evaluated in an in vitro BBB consisting of immortalized adult rat brain microvascular endothelial cells (SV-ARBEC) grown with a rat-conditioned medium on a semipermeable membrane. FC5, FC44, EG2 (a VHH targeting EGFR), and A20.1 (a VHH targeting *Clostridium difficile* toxin A) were co-administered to the top chamber (20 µg/mL each) [136] (Table 7). Then the transmigrated VHHs were quantified at 5, 30, and 60 min by a highly sensitive and specific mass spectrometry-based method: the multiple reaction monitoring (MRM or SRM)–isotype labeled internal standards (ILIS). A time-dependent accumulation of FC5 and FC44 in the bottom chamber was observed, whereas EG2 could not be detected, and only a small amount of A20.1, which did not accumulate over time, was obtained. FC5 was the first detected at 15 min. It ended up with the highest accumulation level at 60 min [136].

Using a similar experimental procedure, a VHH recognizing IGF-1R, which shows a 3-fold higher apparent permeability coefficient than FC5, was identified by Stanimirovic et al. [137] (Table 7).

In addition, a VHH E9 targeting cytosolic human GFAP was reported for its ability to cross a monolayer of immortalized human cerebral microvascular endothelial cells/D3 (hCMEC/D3) (Table 7). From the 1 µM of E9 VHH added onto the cells, 7.8% were detected by ELISA at 60 min in the lower chamber [138].

BBB in vitro models could be practical for screening, but they have some limitations. It can be illustrated with VHHs found unable to cross the BBB while, after systemic administration, they could actually be detected in the brain parenchyma [136]. Indeed, that is the case with the fluorescently labeled EG2 and A20.1, described above, injected intravenously (two injections, 7 mg/kg each, given 1 h apart), although their brain accumulations remain lower than that of the fluorescently labeled FC5 (Table 7). Note that in this study, we cannot exclude the possible role of a dose-effect relationship in brain accumulation.

There are also cases with the VHHs ni3a and pa2H which were tested with FC5 as a positive control [139]. These three undergo an active transport in vitro with a higher BBB crossing efficiency for ni3a and pa2H (Table 7). However, once tested in animal models, they did not show apparent accumulation on their target, β-amyloid deposits. The absence of the BBB uptake in vivo is hypothesized to be due to a lower dose injected or the absence of the active transporters in the mouse model chosen [139,140]. Thus, the prediction of VHH brain penetration through BBB models does not always provide accurate information on real brain entry.

### 3.2. Detection in Brain Homogenates and Slices

Being aware of the limitations of in vitro models, the capacity of sdAbs to penetrate the brain has been assessed either in the whole brain (Figure 1B) or on brain sections (Figure 1C) after peripheral administration in animals. Regarding the first approach, C57BL/6 mice were injected intravenously with soluble FC5 and FC44 at 30 µg per mouse (approximately 1.7 mg/kg). Four hours later, unbound sdAbs were eliminated by intracardiac perfusion of the animals with saline. Capillaries were depleted, and brain dissections snap-freezed. FC5 and FC44 were then extracted from the brain homogenates by Ni^2+^-affinity purification. Both FC5 and FC44 were detected by western blot [135] (Table 7). The respective concentrations were not determined in that study. Performing ion affinity chromatography to extract the sdAbs is beneficial to clarify the substrate from irrelevant proteins. However, this can lead to a certain loss. sdAbs can be degraded or left trapped in the sample. To limit this loss, one possible way is to inject radiolabeled sdAbs and measure the radioactive emission directly from the total brain homogenate. However, in this case, a controlled number of radioisotopes per sdAb is preferable for correct quantification.

The presence of sdAbs after systemic administration can also be assessed by ex vivo fluorescence imaging on a total perfused brain. The scanning of the intact brain with imaging systems allows a more visual detection of the brain-penetrated sdAbs. This was tested with VHHs coupled with a near-infrared fluorescent imaging probe NHS-IR800 (LiCor, Lincoln, NE, USA), which covalently binds to the primary amines of the protein (N-terminus and/or accessible lysine side chains) [136]. To obtain more details on brain biodistribution, studies can be done on brain sections, as with the VHHs FC5-IR800, EG2-IR800, and A20.1-IR800 (two consecutive injections of 7 mg/kg given 1 h apart) [136] (Table 7). Apart from fluorescence, autoradiography is also possible. It is more sensitive and reduces the background signal that one can have because of the tissue’s autofluorescence. However, this technique may require a long-time exposure to the radiolabeled sdAb to obtain results (around a three-week exposition) [141].

However, chemical coupling can trigger a loss of binding affinity and/or functional activity, as observed with FC44-IR800 [136]. It may alter the BBB crossing efficiency or the brain accumulation of the VHH and then compromise the study. The same goes for radiolabeling.

To avoid this chemical coupling, one way is to perform immunofluorescence or immunohistochemistry on the brain slices with an anti-VHH antibody or directed against a tag present in the sdAb construct (such as histidine, cMyc or Flag tags). To demonstrate the capacity of VHH E9 to reach its cytosolic astrocytic target GFAP, mice were perfused with 400 µg, 4 mg, or 25 mg of E9 for 60 min via the carotid artery to limit plasma clearance. Clear immunostaining of astrocytes was obtained with 4 mg and 25 mg 1 h after the intracarotid infusion. Staining was observed intensely in the ipsilateral hemisphere in the astrocytic endfeet adjacent to blood vessels (Table 7). It was also observed in the corpus of astrocytes located in several regions: the corpus callosum, the hippocampus, the olfactory bulb, and the gray matter. When injected intravenously (2 mg), the immunostaining was specifically observed in the proximity of ventricular regions. This confirmed the in vivo brain uptake of E9 VHH. It also demonstrated that it can still reach its cytoplasmic target and acts like an intracellular antibody fragment (also called intrabody) [138].

### 3.3. Quantification in Brain Fluids

Molecules administered systemically can enter brain parenchyma by reaching the cerebrospinal fluid (CSF) first via passage across the blood-CSF barrier. They can cross the fenestrated capillaries (60–80 nm fenestrations [142]) and the surrounding epithelial cell monolayers that form the choroid plexus and other circumventricular organs [143]. However, the amount of this passage is limited by the relatively low surface exchange compared to that of the BBB. Thus, the main brain route from the blood is crossing the BBB. The molecules are then transported by diffusion or convection from the interstitial fluid (ISF) to the CSF (for review see [143]). Whether it is directly through the choroid plexus or indirectly through the BBB, at a certain point, molecules end up in the CSF. They move along with CSF’s flow throughout the CNS: from lateral and third ventricles to the fourth ventricles; then into the cerebral and spinal subarachnoid spaces; and finally, undergo a re-absorption into venous blood. Thus, brain-penetrating molecules can be detected in the CSF compartment. Given that the precise site of action of a neurotherapeutic is likely unknown and that the accumulation site could rather correspond to a sequestration zone, CSF was thought to be a good surrogate. It is not surprising that many studies have used CSF concentration as an indicator of drug brain access. Traditional concentration-effect studies using lipophilic compounds provided evidence that supports CSF concentration as a reference for examining the pharmacodynamic characteristic of a pharmacological agent. The sampling is practical. It can easily be performed through catheters inserted in the cisterna magna (Figure 1D) [143]. However, the experimenter has to be careful not to compromise the sample with blood contamination. This approach was used by Haqqani and colleagues in 2013 to quantify the brain delivery of four VHHs: FC5, FC44, EG2, and A20.1, described above (Table 7). They were co-administered in 3 consecutive injections of 7 mg/kg given 1 h apart. The CSF was collected 15 min after the last injection. The sensitivity of liquid chromatography coupled with mass spectrometry method (NanoLC-SRM-ILIS) allowed the detection of unlabeled nanobodies at 1.7 ng/mL without removing proteins naturally abundant in the CSF [136].

However, opinions differ within the scientific community concerning the relevance of using the antibody or sdAb’s concentration in the CSF in assessing its entry into the brain. Indeed, some argue that all blood proteins can be found in a size-dependent manner in the CSF due to the permeability of the choroid plexus. Therefore, this might not be evidence of transport into the brain, unlike a direct quantification from the brain interstitial space [144]. Such a detection way could be possible with the development of cerebral microdialysis (Figure 1D) (Table 7). Initially used to collect free monoamines and other small molecules from neural tissues, this technique is being extended to macromolecule samplings such as sdAbs and IgGs [145,146,147]. This is due to the recent development of probes with larger pore membranes and an adapted push−pull system [148,149,150]. In cerebral microdialysis experiments, it is important that the probe implantation does not significantly alter the integrity of the BBB. This could compromise the concentration values. Thus, a delay of 16–24 h after probe implantation is given before starting the experimental procedure to allow the BBB to recover. The integrity of the BBB can be inspected by the extravasation of Evans Blue or FITC-dextran in the vicinity of the probe by using fluorescence microscopy [145,146].

To investigate the brain uptake of monovalent nanobodies, the concentration of the VHH An-33 against the *Trypanosoma brucei brucei* variant-specific surface glycoprotein was determined in hippocampal microdialysates of both healthy rats and rats with encephalitic stages of African trypanosomiasis [145] (Table 7). The ELISA analysis of unlabeled Nb An-33 administered intravenously (4 mg/kg bolus) showed only a limited proportion (~0.0005%) of the injected dose, corresponding to 50 ng/mL. This concentration is significantly below the therapeutic concentration needed (≥0.5 µg/mL), as determined in vitro.

In another study, the concentration of VHHs from hippocampal microdialysates was determined for a bispecific nanobody (Nb105) consisting of an anti-transferrin nanobody (Nb62) fused to a nanobody targeting the green fluorescent protein (GFP) (Table 7). The same anti-GFP nanobody fused to an irrelevant nanobody raised against a chicken lysozyme was used as a negative control. Mice received a first dose of 250 pmol/g of either Nb105 or control intraperitoneally, followed by a higher dose (750 pmol/g) 240 min later. The quantification by a wash-free assay named AlphaScreen™ showed a peak concentration of Nb105 at 150 min. After the second injection, the negative control bispecific nanobody was also detected. This could be explained by a partial recovery of the BBB. It could also support the idea that circulating macromolecules can nonspecifically enter through the choroid plexus. This observation was confirmed by a recent in vivo sampling technique, cerebral open-flow microperfusion (Figure 1D and Table 7). It is similar to microdialysis, except the probe has a 100 µm-wide open exchange area instead of a membrane. With this technique, the delay between probe implantation and sampling could be prolonged to 14 days, which gave more time for BBB recovery [146].

Moving from relative detection to absolute quantification in a brain parenchyma region can help to predict the therapeutic effectiveness of an antibody candidate and adapt the dosage. As discussed by Shen et al. (2004) [143], based on data from a dozen published preclinical pharmacokinetic studies on small molecule drugs (antibiotics, analgesics, antidepressants, anticancer and antiepileptics), the CSF-to-ISF ratio may be greater than 1 (CSF > ISF) or the expected less than 1 (CSF < ISF). The first case could be explained by a favored efflux transport at the BBB or the choroid plexus. It could also be attributed to active intracellular uptake or sequestration. In the second case, the lower CSF concentration is consistent with a sink action of CSF. It could also reflect a slower kinetic equilibration in the CSF space. The determination of ISF concentration appears to be more relevant in both cases since the CSF level would overestimate or underestimate the actual concentration of the candidate molecule in the site of action. However, the molecule partitioning among the different brain compartments can be complex. Regional variation in brain ISF and along the CSF flow path has been reported. To our knowledge, no published study has investigated the pharmacokinetics of sdAbs in different brain regions in a simultaneous manner. However, there is a study on endogenous rat IgG and the exogenous human-specific antibody trastuzumab that tends to confirm this difference in brain compartment distribution. At steady-state, the endogenous IgG is more concentrated in the cisterna magna CSF (CSF_CM_) than in the lateral ventricle CSF (CSF_LV_), while the concentration in the CSF_LV_ is similar to the one in the striatum ISF (ISF_ST_). Regarding the antibody trastuzumab, which has no target in rats, CSF_LV_ concentration was found to be higher compared to that of the ISF_ST_ at the initial time points following its systemic administration. This suggests a first entrance through the choroid plexus. On average, the CSF_CM_ and CSF_LV_ were not statistically different. At the same time, CSF_CM_ was slightly lower than IST_ST,_ which is the opposite of the previous findings on endogenous IgG. These observations provide insights into the importance of the sampling site.

### 3.4. Evaluation in Living Animals

Imaging techniques, such as positron emission tomography (PET) and single photon emission computed tomography (SPECT), are extensively used in clinical neuroscience. They significantly contributed to the understanding of the cellular and molecular mechanisms underlying CNS pathophysiology. For preclinical research, these technologies have been adapted for small laboratory animal models. It gave the possibility to visualize in real-time the brain uptake of radiolabeled VHHs as demonstrated by Lesniak and colleagues by PET imaging [151] (Figure 1E). The authors showed that a [^89^Zr]nanobody without a specific target in mouse brain elicits a negligible brain-uptake upon intravenous administration in comparison with an intracarotid route (Table 7). The intracarotid infusion resulted in a peak concentration of 25.79 ± 15.79 %ID/cc (% of injected dose per cubic centimeter of tissue) in the ipsilateral hemisphere. However, no accumulation was observed in the contralateral one. At 24 h post-infusion, half of the nanobody still remained. In addition to that study, we can also mention the μ-SPECT and PET analyses of [^99m^Tc]VHH An-33 and [^125^I]VHH-E9, respectively [141,145] (Table 7).

Selecting a suitable physiological readout is a key to helping assess the potential of a candidate sdAbs to validate both the brain penetration and the therapeutic value of a candidate sdAb. It is provided by physiological readouts which are known to be modulated by the intracerebral target (Figure 1E) (Table 7). It can be hypersensitive to thermal stimuli, as observed in inflammatory pain models such as the Hargreaves model. The paw withdrawal latency is reduced in sick animals because of thermal hyperalgesia. A single intravenous injection of FC5 (7 mg/kg) does not increase the latency [137]. Neither does 1 mg/kg of galanin, a neuropeptide that is known to produce analgesia by binding to GalR1 and GalR2 receptors expressed in the brain. This means that FC5 has no effect on thermal analgesia, and that galanin cannot cross the BBB on its own when given systemically. Interestingly, galanin chemically conjugates to FC5 (6 mg/kg) or to an IGF1R-targeting VHH (IGF1R-4) (3 mg/kg), exhibiting an analgesic effect. It is 4-fold more pronounced with IGF1R-4-galanin, suggesting a better brain uptake of IGF1R-4 in comparison with FC5 [137].

In the same pain model, the brain penetration of FC5 fused with the human Fc domain was evaluated by using dalagrin and neuropeptide Y as BBB-impermeable neuropeptides [152]. This approach could be tested on VHH targeting G-protein coupled receptors (GPCRs) involved in pain alleviation, such as the metabotropic glutamate receptor type 1. Inhibiting its function with an engineered BBB-crossing antibody triggered an analgesic effect on the Hargreaves model [153]. It could be interesting to test unmodified sdAbs. The large group of GPCRs represents more than 30% of therapeutic drug targets and still constitutes a niche for future drug discovery [154] (Table 7).

Monitoring body temperature can also be a good readout to assess brain delivery. As an example, the work of Wouters and his colleagues in 2020 on an anti-transferrin receptor nanobody Nb62, described above, fused with neurotensin [155]. Neurotensin is a BBB-impermeable neuropeptide, which elicits hypothermia when present in the brain. A body temperature drop was observed in mice that received a dose of 250 pmol/g intraperitoneally. A peak was obtained 150 min after the injection. In contrast, no functional response was observed with the negative control VHH (Table 7).

Of note, the choice of readout can impact the delay in obtaining results or its relevance for the therapeutic purpose of the sdAb. For instance, cognitive-related behavior will require a more complex and longer experimental procedure.

## 4. Conclusions

In summary, immunotherapy continues to grow in the field of neurology. Among the antibodies, sdAbs are a promising breakthrough in the therapeutic field as they have a better capacity to penetrate the brain. Depending on the pathology, the integrity of the BBB can be modified, facilitating more or less this brain diffusion. Upon systemic administration, sdAbs are able to cross the BBB without any additional strategy (BBB disruption or chemical coupling or modifications), even though the mechanisms of entry into the brain are still unclear for some. To help predict rapidly promising sdAb drug candidates for brain disorders, the development of sensitive and adequate detection methods in the brain may be crucial. Hence, investigations on a good technique in terms of short completion time, high sensitivity, and relevance of models are progressing (Table 7).

**Table 7 ijms-24-02632-t007:** Nonexhaustive list of techniques used to assess brain penetration of sdAbs.

Evaluation Method	Protocol	Assay Used	sdAb Evaluated	sdAb Format	Target	Key Findings	References
Transmigration across in vitro BBB model	HCEC + medium conditioned by fetal human astrocytes	ELISA	* FC5	VHH	TMEM-30A	Both FC5 and FC44 cross the in vitro BBB in an energy-dependent manner	[135]
			** FC44		HCEC proteins		
	SV-ARBEC	Mass spectrometry (MRM-ILIS)	FC5		TMEM-30A	FC5 crosses more rapidly than FC44 and A20.1. A20.1 did not accumulate over time while FC5 reached the highest accumulation level at 60 min time point	[136]
			FC44		HCEC proteins		
			A20.1		Clostridium difficile toxin A		
	SV-ARBEC		IGF1R3, IGF1R4, IGF1R5		Extracellular domain of the human IGF-1R	The three VHHs cross the in vitro BBB, while A20.1 cannot. IGF1R3 shows a 3-fold higher apparent permeability value compared to FC5	[137,156]
	HCMEC/D3	ELISA	E9		Cytosolic human GFAP	7.8% of the applied quantity of E9 was detected at 60 min time point	[138]
	BCEC + newborn rat astrocytes		ni3a, pa2H		Aβ brain aggregates	VHHs cross the in vitro BBB in an energy-dependent manner and with a higher transmigration velocity than FC5	[139,140]
Ex vivo detection after peripheral administration	i.v. administration + brain homogenization	Western blot	FC5	VHH	TMEM-30A	Both FC5 and FC44 could be detected by western blot after their extraction from brain homogenates by ion affinity chromatography	[135]
			FC44		HCEC proteins		
	i.v. administration + optical tomography sectioning of brain	Fluorescence imaging	FC5		TMEM-30A	FC5-injected animals showed higher fluorescence in the brain compared to control VHHs (EG2 and A20.1)	[136]
	intracarotid infusion or i.v. administration + brain slicing	Immunostaining	E9		Cytosolic human GFAP	Astrocytes were immunostained at 1 h post-intracarotid injection (4 and 25 mg/kg). An administration by i.v. route showed immunostaining in the proximity of ventricular regions	[138]
	i.v. administration in P2APP mice + brain slicing	Fluorescence imaging	R3VQ		Aβ brain aggregates	Amyloid plaques were labeled throughout the brain 4 h post-injection	[103]
	i.v. administration in Tg4510 + brain slicing		A2		Tau inclusions	Neurofibrillary tangle-like structures were stained throughout the brain 4 h post-injection	[103]
	i.p. administration in FVB/N mice + brain homogenization or brain slicing	ELISA and immunohistochemistry	PrioV3		Isoform scrapie prion protein PrPSc	The immunodetection showed a biphasic pattern in brain homogenates. PrioV3 accumulated in the hippocampus, the alveus, and the cerebellar cortex from 4 to 24 h post-injection	[134]
	i.v. administration in wild type and and ArcSwe mice	Autoradiography	E9		Cytosolic human GFAP	The radiolabeling signal was more intense throughout the brain of ArcSwe mice at 8 h, but did not show a clear association with the target (GFAP) staining	[141]
	i.v. administration + brain homogenization or brain slicing	ELISA and immunohistochemistry	*** TXB2 fused with human Fc domain	Fc-fused VNAR	Transferrin receptor	A brain concentration of 6 nM was found at 18 h. Immunoreactivity was observed in endothelial cells, choroid plexus epithelial cells and neurons in different regions of the brain	[157]
	i.v. administration in a mouse model of brain cancer + brain slicing	Fluorescence imaging	VH-9.7	VH	Glioblastoma stem-like cells (GSC)	VH-9.7 was localized to the human 22 GSC orthotopic xenografts	[62,64]
Detection in brain fluids after peripheral administration	i.v. administration in rats + CSF sampling	Mass spectrometry (NanoLC -MRM-ILIS)	FC5	VHH	TMEM-30A	The unlabeled VHHs could be detected at very low amount (1.7 ng/mL) without removing proteins naturally occurring in the matrix.	[136]
			FC44		HCEC proteins		
			EG.2		EGFR		
			A20.1		Clostridium difficile toxin A		
	i.v. administration in both healthy and rats with encephalitis + hippocampal microdialysate sampling	ELISA + radioactivity measurement	An-33		Trypanosoma *brucei brucei* variant-specific surface glycoprotein	Approximately 0.0005% of the administered dose was detected, which is below the therapeutic concentration	[145]
	i.p. administration in mice + followed by microdialysate collection	Alpha-Screen	Nb105	bi-VHH	Transferrin receptor and green fluorescent protein	A peak concentration was observed at 150 min time point	[146]
	i.v. administration or intracarotid infusion in healthy C3HeB/FeJ mice	PET/CT	Nb11		No specific target in mouse brain	The nanobody showed a higher brain-uptake via intracarotid route but did not accumulate in the controlateral side. Half of the nanobody still remained at 24 h	[151]
	i.v. administration in both healthy and rats with encephalitis	μ-SPECT	An-33		*Trypanosoma brucei brucei* variant-specific surface glycoprotein	Only a small portion of the VHH reaches the brain parenchyma in healthy animals (∼0.0005% of the initial dose). This passage is increased in the pathological condition	[145]
	i.v. administration in wild-type and ArcSwe mice	PET	E9		Cytosolic human GFAP	VHH E9 displayed a brain average concentration of 0.15% ID/g at 2 h post-injection in wild-type mice. The radiolabeling was detected in the brains of both wild-type and ArcSwe mice up to 24 h	[141]
	i.v. administration in P2APP mice	*in vivo* two-photon imaging	R3VQ		Aβ brain aggregates	Both plaques and vascular Aβ were stained in the cortical surface up to 350 μm deep at 30 min post-injection. It persisted up to 4 h post-injection	[103]
	i.v. administration in Tg4510		A2		Tau inclusions	Neurofibrillary tangle-like structures were stained in the cortical surface up to 350 μm at 2 h post-injection. It persisted up to 4 h post-injection	[133]
	i.v. bolus injection in wild-type and APP/PS1 transgenic mice	PET/SPECT	ni3a, pa2H		Aβ brain aggregates	Both ni3a and pa2H show lower brain uptake than FC5	[140]
Physiological readout	i.v. administration of the sdAb fused with galanin in the Hargreaves model	Analgesic effect	FC5	VHH	TMEM-30A	IGF1R4-galanin exhibited a more pronounced analgesic effect than FC5-galanin. This suggests a better uptake of IGF1R4	[137]
			IGF1R4		IGF-1R		
	i.v. administration of the sdAb fused with galanin or neuropeptide Y in the Hargreaves model		FC5 fused with human Fc domain	FC-fused VHH	TMEM-30A	Systemic administration of FC5-dalagrin induced an analgesic response with a maximal effect obtained after three injections of 7 mg/kg separated 1h apart. Systemic dosing of FC5-neuropeptide Y suppressed thermal hyperalgesia	[152]
	i.v., i.p. and s.c. administration of the sdAb fused with neurotensin in TLR4^-/-^ mice	Hypothermia	Nb62	VHH	Transferrin receptor	A peak effect was obtained at 100–110 min and 120–180 min after i.v. and i.p. injections, respectively. The drop amplitude is doubled by the i.p. route (–6 °C) and the hypothermic effect lasted 7 h compared to 3 h by i.v. route. The subcutaneous route showed the more prolonged effect	[155]
	i.v. administration of the sdAb fused with neurotensin in mice	Hypothermia	*** TXB2 fused with human Fc domain	Fc-fused VNAR	Transferrin receptor	TXB2-hFc induces a drop in temperature in a dose-dependent manner at 2 h time point and returned to normal by 6 h. The minimal dose required to produce hypothermic effect is 10 nmol/kg (0.75 mg/kg)	[157]

* GenBank no. AF441486. ** GenBank no. AF441487. *** VNAR TXB2 fused to human IgG1 Fc likely corresponds to TXB4.

## Figures and Tables

**Figure 1 ijms-24-02632-f001:**
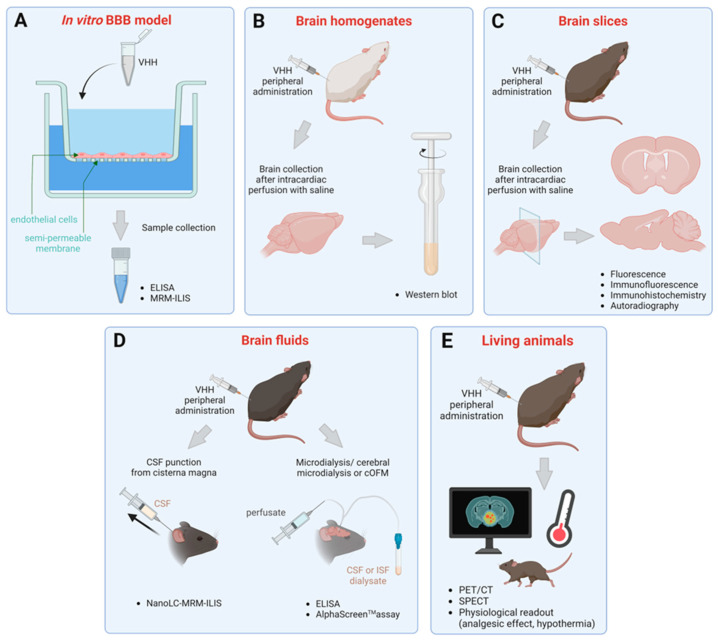
A schematic representation of the methods previously reported to detect and quantify the passage of the BBB of single-domain antibodies. (**A**) Brain uptake measurement using in vitro endothelial cell layers-based BBB model followed by ELISA or MRM-ILIS assays. (**B**) Detection in brain homogenates by western blot after peripheral administration in animal. (**C**) Detection on brain slices using optical microscopy or autoradiography after peripheral administration in animal. (**D**) Quantification in punctioned brain fluids using ELISA, or a highly sensitive mass spectrometry method (NanoLC-MRM-ILIS), or by an energy transfer-based assay (AlphaScreen^TM^), after peripheral administration in animal. (**E**) Brain entry after peripheral administration assessed by a real-time PET/CT, SPECT or different physiological readouts in living animals. Created with BioRender.com (accessed on 2 November 2022).

**Table 1 ijms-24-02632-t001:** Selected examples of antibodies used for multiple sclerosis. i.v., intravenous; s.c., subcutaneous.

Antibody	Target	Clinical Status	Dose	Key Findings/Mode of Action	References
Natalizumab (Tysabri^®^) *Humanized mAb*	α4β1 integrin	Approved by FDA and EMA	i.v. infusion of 300 mg every 4 weeks	Blocks binding of α4β1 integrin to VCAM	[26]
Rituximab (Rituxan^TM^) *Chimeric mAb*	CD20	Phase III	i.v. infusion of 500 or 1000 mg every 6–12 months	Reduced recurrences by lysing circulating B cells and stops MS inflammation	[28,29]
Ocrelizumab (Ocrevus^TM^) *Humanized mAb*		Approved by FDA and EMA	i.v. infusion of 300 mg day 1 and 5 and 600 mg every 6 months		[30]
Ofatumumab (Kesimpta^®^) *Humanized mAb*		Approved by FDA and EMA	s.c. injection 20 mg on weeks 0, 1, and 2, and 20 mg each month		[40]
Ublituximab (TG-1101) *Chimeric IgG1 mAb*		Phase III Pending FDA approval	i.v. infusion of 150 mg day 1, 450 mg day 15 and 450 mg every 6 months		[31]
Alemtuzumab (Lemtrada^TM^) *Humanized mAb*	CD52	Approved by FDA and EMA	i.v. infusion 12 mg/day for 5 days and 1 year later 12 mg/day for 3 days	Lysis of T and B lymphocytes	[32]
Opicinumab (BIIB033) *Human mAb*	LINGO-1	Phase II Development stopped	i.v. infusion 750 mg every 4 weeks for 96 weeks	Inhibition of LINGO-1, differentiation of oligodendrocyte precursor cells into mature oligodendrocytes allowing remyelination	[33]
Elezanumab (ABT-555) *Human mAb*	RMGa	Phase II	i.v. infusion of 1.800 mg monthly or bimonthly	Inhibition of RMGa promoting regeneration	[34]
Temelimab (GNbAC1) *Humanized IgG4 mAb*	MSRV-Env protein	Phase II	i.v. infusion of a single dose of 36, 60, 85 or 110 mg/kg	Expected inhibition of MSRV-Env decreasing proinflammatory and autoimmune cascades	[35]

**Table 2 ijms-24-02632-t002:** Approved therapeutic antibodies for migraine.

Antibody	Target	Clinical Status	Dose	Key Findings/Mode of Action	References
Erenumab (Aimovig^®^) *Human mAb*	CGRP receptor	Approved by FDA and EMA	s.c. injection of 70 mg once per month	Blocks the CGRP receptor and inhibits the activity of CGRP, preventing the onset of pain	[53]
Fremanezumab (Ajovy^TM^) *Humanized IgG2 mAb*	α and β isoforms of CGRP		s.c. injection of 225 mg once per month or 675 mg every 3 months	Targets the CGRP ligand and blocks its binding to the receptor	[51]
Galcanezumab (Emgality^®^) *Humanized mAb*			s.c. injection of 120 mg once per month		[50,54,55]
Eptinezumab (Vyepty^TM^) *Humanized IgG1 mAb*			Infusion of 100 mg every 3 months		[49,56]

**Table 5 ijms-24-02632-t005:** Monoclonal antibodies tested for Parkinson’s disease.

Antibody	Target	Clinical Status	Dose	Key Findings/Mode of Action	References
Prasinezumab (PRX002) *Humanized IgG1 mAb*	Aggregated α-synuclein	Phase II	i.v. infusion of 1500 or 4500 mg every 4 weeks for 52 weeks	Trend toward benefit in motor functions	[105]
Cinpanemab (BIIB054) *Human-derived mAb*	Aggregated α-synuclein	Phase II—discontinued	i.v. infusion of 250, 1250 or 2500 mg every 4 weeks for 52 weeks	No improvement in motor functions	[108]
MEDI1341 (TAK-341) *mAb*	Monomeric and aggregated α-synuclein	Phase I	i.v. infusion of 3 doses given at 4 weeks interval	No results released	/
PFFNB2 *sdAb*	α-synuclein preformed fibrils	Preclinical stages	Intraventricular injection of AAV encoding PFFNB2 in the transgenic mouse model PACTg(SNCAWT)	In vitro dissociation of fibrils. Prevents the spreading of pathological α-synuclein to the cortex	[114]
NbSyn2 *sdAb*	Monomeric and fibrillar α-synuclein		/	Inhibit the formation of fibrils and converts toxic oligomers into less toxic species	[115,116,117,118,119,120]
NbSyn87 *sdAb*			Injection in the substantia nigra of AAV encoding NbSyn87 into rats		

**Table 6 ijms-24-02632-t006:** Tested antibodies for Creutzfeldt-Jakob’s disease. i.p., intraperitoneal.

Antibody	Target	Clinical Status	Dose	Key Findings/Mode of Action	References
PRN100 *Humanized IgG4κ mAb*	PrP^Sc^	No clinical trial—“special exemption”	i.v. infusion of 80–120 mg/kg every 2 weeks until death	Treatment is safe and reached CSF and brain tissue concentrations expected	[133]
PrioV3 *sdAb*		Preclinical stages	i.p. injection of 5 mg/kg in WT mice	Crosses the BBB	[134]

## Data Availability

Not applicable.

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
