# Peer review of "Targeting the Brain with Single-Domain Antibodies: Greater Potential Than Stated So Far?"

_ijms, 2023, doi:10.3390/ijms24032632_

Round 1

Reviewer 1 Report

Congratulations, your manuscript is well-developed and structured.

  1. What is the main question addressed by the research?

This review aims to illustrate the progressive use of mAbs in neurodegenerative diseases and neurooncology therapy and the promising applications of sdAbs in the same field. Second, the authors describe the methods used to assess the ability of the sdAbs to enter the brain parenchyma, with a particular emphasis on new rapid and efficient technical approaches to accelerate and improve the screening and development of single-chain domains.

  1. Do you consider the topic original or relevant in the field? Does it address a specific gap in the field?

Single Domain Antibodies are small molecules with modular structure and ease of expression. Single domain antibodies (sdAb) have a wide range of applications, including as an analytical design tool, and are, therefore, of great interest to synthetic biologists and bioengineers.

3. What does it add to the subject area compared with other published material?

The manuscript is essential and enables effective use and sharing of existing sdAbs related to brain entrance, including those with engineered functions (e.g., fusions with fluorescent proteins), as well as the rational design and engineering of new sdAbs.

4. What specific improvements should the authors consider regarding the methodology? What further controls should be considered?

The authors summarized the information and experimental data regarding its biogenesis and the possible mechanisms to cross the blood-brain barrier. They developed an excellent analysis of publicly available, sdAb-focused information, providing sdAb data from protein databases and published scientific articles.

5. Are the conclusions consistent with the evidence and arguments presented and do they address the main question posed?

The authors reviewed sdAbs or antibody fragments, which have shown promise in vivo by their ability to reach the brain tissue upon systemic administration without any additional strategy, such as BBB disruption or chemical coupling or modifications. After its analysis, they provided a path to help predict rapidly promising sdAb drug candidates, indicating the importance of developing sensitive and adequate detection methods in the brain for neurological diseases.

6. Are the references appropriate?

Yes. All the references reported are well indicated.

7. Please include any additional comments on the tables and figures.

The figure and the tables are well presented.

Author Response

Congratulations, your manuscript is well-developed and structured.

We would like to thank the reviewer for his/her positive comment.

English language and style are fine/minor spell check required.

We checked the spelling and the grammar.

Reviewer 2 Report

In this manuscript, the authors provide an overview of methods used to assess or evaluate brain penetration of single-domain antibodies (sdAbs) and discuss the pros and cons that could affect the identification of brain-penetrating sdAbs of therapeutic or diagnostic interest.

Please summarize the evidence supporting application of antibodies in brain disease in a table/figure in section 2.

Please summarize techniques to assess brain penetration of sdAbs in animals and humans in a table. This information will be valuable for researchers in the field. 

Please summarize key findings on therapeutic or diagnostic antibodies that cross the BBB in brain disorders in a table. 

Author Response

Please summarize the evidence supporting application of antibodies in brain disease in a table/figure in section 2.

We agree that a table that summarizes the antibodies mentioned in the manuscript would help to better understand and to pinpoint the main information. Thus, we added six different tables, one table for each pathology: Table 1 can be found in page 3 - line 98; Table 2 in page 4 - line 136; Table 3 in page 5 - line 176; Table 4 in page 7-line 249; Table 5 in page 9 – line 289, and Table 6 in page 10 – line 337.  Each table shows the following points for one specific brain disease: the name of the antibodies tested (medical name) with the nature of the isotype (humanized, chimeric, single-domain antibody), their targets, their clinical status (preclinical, clinical stage, authorized), the dosing, the key findings related to their clinical status or their mode of action, and the references.

Please summarize techniques to assess brain penetration of sdAbs in animals and humans in a table. This information will be valuable for researchers in the field.

We agree with the reviewer. Thus, we added a Table 7 in page 18 – line 628. It contains the following points: the evaluation method corresponding to the methods used for sampling, the protocol (cells used to constitute the in vitro BBB, the route of administration), the assays used to detect the sdAbs in the samples, the name of the sdAbs tested, their format (VHH, VH, VNAR), their target, the key findings and the references.

Please summarize key findings on therapeutic or diagnostic antibodies that cross the BBB in brain disorders in a table.

We agree with the reviewer. Thus, we added this information for the therapeutic antibodies in the seven tables previously mentioned (Table 1 – 7). Also, the majority of the mentioned antibodies (unlike the mentioned sdAbs) act outside the brain or in region where the BBB is not present. Therefore, we cannot say that these are BBB-crossing antibodies. This point has already been specified in the manuscript (see paragraph 4, Conclusion).

English language.

We checked the spelling and the grammar.

Reviewer 3 Report

Tsitokana et al present an interesting review article on the use of Nanobodies (VHH) and variable domains of antigen receptors (VNAR) for the treatment of neurological diseases. The topic is quite interesting for a broader public but stays superficial in larger parts of the manuscript. A molecular description of VHH and VNAR themselves and their functional mechanisms is missing. Instead neurological diseases have been described in which the putative medications have been applied. However, these diseases are described in a very superficial and therefore inappropriate manner. Some technical details on the other hand in the second half of the manuscript, e.g., 3.2, are described in great detail. This renders the manuscript quite unbalanced in total. Eventually, it remains unclear whether or not VHH and VNAR are promising new therapies in neurological diseases.

Specific points:

-        Various VHH and VNAR are named within the manuscript either with individual name, medical name or there clone abbreviations. Occasionally, target molecules they recognize are described but often not. Some are in clinical phases, some not. Which ones are real medications? In total this makes the text hard to read therefore one or more table(s) listing the VHH/VNAR, the target it recognizes, the trial phase it is currently tested and some additional information would be highly appreciated.

-        The authors argue that the VHH and VNAR do not enter the brain due to the BBB. This is only partially true as various neurological diseases coincide with a breakdown of the blood-brain barrier (BBB), e.g., multiple sclerosis. It depends on the type of disease and neoplasia may dispaly both, a strengthening or a breakdown of the BBB in different regions of the tumor. The manuscript should reflect this in a more careful manner.

-        Page 4, line 185: “brain progression”?

-        Page 10, lines 434+435: The authors argue that macromolecules enter the cerebrospinal fluid due the fenestrated endothelium of the choroid plexus. This is anatomically incorrect and would be disastrous for the organism. While it is correct that the choroid plexus counts to the circumventricular organs and is fenestrated it is covered by an epithelium that forms the blood-liquor barrier. Under physiological conditions the liquor cerebrospinalis is an ultrafiltrate of the blood serum, contains a small amount of cells and is low in proteins. In no case it is widely open as suggested in the manuscript.

-        Page 12, line 582 “prerequisite” instead of “requisite”?

Author Response

A molecular description of VHH and VNAR themselves and their functional mechanisms is missing.

We understand your point of view; however, we would like to focus the content of our manuscript straight to their use in brain disorders. However, we added the following sentence: "Their molecular features and functional mechanisms have been well described in the literature [14–19]" (page 1, lines 39-40).

Instead neurological diseases have been described in which the putative medications have been applied. However, these diseases are described in a very superficial and therefore inappropriate manner. Some technical details on the other hand in the second half of the manuscript, e.g., 3.2, are described in great detail. This renders the manuscript quite unbalanced in total.

We agree with the reviewer’s comments but we think that adding more details regarding the different pathophysiologies would overwhelm the readers. They could miss the main objectives of this review, which are doing a state of the art of the use of antibodies and sdAbs in brain diseases and discuss the techniques used to evaluate brain penetration of sdAbs in particular.

Eventually, it remains unclear whether or not VHH and VNAR are promising new therapies in neurological diseases

To better highlight the increased potential of sdAbs as therapeutics for brain diseases, we have greatly shorthened the Discussion.

Various VHH and VNAR are named within the manuscript either with individual name, medical name or their clone abbreviations. Occasionally, target molecules they recognize are described but often not. Some are in clinical phases, some not. Which ones are real medications? In total this makes the text hard to read therefore one or more table(s) listing the VHH/VNAR, the target it recognizes, the trial phase it is currently tested and some additional information would be highly appreciated.

We totally agree with the reviewer. To our knowledge, the most advanced VHHs and VNARs tested for neurological diseases are in preclinical stages. There is only one VHH (caplacizumab – Cablivi®) that has been approved by FDA and currently used as a medication.

As suggested by the reviewer, we added the information on the targets, the status of the VHHs/VNARs/VH in the six new tables (Table 1-6).

The authors argue that the VHH and VNAR do not enter the brain due to the BBB. This is only partially true as various neurological diseases coincide with a breakdown of the bloodbrain barrier (BBB), e.g., multiple sclerosis. It depends on the type of disease and neoplasia may dispaly both, a strengthening or a breakdown of the BBB in different regions of the tumor. The manuscript should reflect this in a more careful manner.

This point raised by the Reviewer is important. Thus, we added the following sentence in the conclusion: “Depending on the pathology, the integrity of the BBB can be modified, facilitating more or less this brain diffusion” (page 16, lines 603-604).

Page 4, line 185: “brain progression”?

We rectified into “AD progression” (page 6 - line 195). We also modified “brain progression” elsewhere into "the tumor progression in brain" (page 5 - line 173).

Page 10, lines 434 to 435: The authors argue that macromolecules enter the cerebrospinal fluid due the fenestrated endothelium of the choroid plexus. This is anatomically incorrect and would be disastrous for the organism. While it is correct that the choroid plexus counts to the circumventricular organs and is fenestrated it is covered by an epithelium that forms the blood-liquor barrier. Under physiological conditions the liquor cerebrospinalis is an ultrafiltrate of the blood serum, contains a small amount of cells and is low in proteins. In no case it is widely open as suggested in the manuscript.

We fully agree with the reviewer. Indeed, it is not widely open as it may have been suggested in our text since there is an epithelium separating the fenestrated capillaries and the CSF and acts as a barrier. Therefore, we modified the sentence (page 14, lines 459-462) as following: “Molecules administered systemically can enter brain parenchyma by reaching the cerebrospinal fluid (CSF) first via passage across the blood-CSF barrier. They can cross the fenestrated capillaries (60-80 nm fenestrations [142]) and the surrounding epithelial cell monolayers that form the choroid plexus and other circumventricular organs [143]”.

Page 12, line 582 “prerequisite” instead of “requisite”?

This word has been deleted in the revised manuscript.

English language.

We checked the spelling and the grammar.